# Analysis of Phenotypic Diversity and Comprehensive Evaluation of 51 *Helleborus* L. Hybrid Individuals

**DOI:** 10.3390/plants14203226

**Published:** 2025-10-20

**Authors:** Liuqing Qu, Bingyu Yuan, Xiaohui Wen, Jia Guo, Jianrang Luo, Xiaohua Shi

**Affiliations:** 1Zhejiang Institute of Landscape Plants and Flowers, Zhejiang Academy of Agricultural Sciences, Hangzhou 311200, China; quliuqing123@163.com (L.Q.); yuanbingyu@163.com (B.Y.); wenxiaohui119@126.com (X.W.); 2College of Landscape Architecture and Arts, Northwest A&F University, Yangling 712100, China; luojianrang@nwafu.edu.cn; 3Zhejiang Xiaojian Group Co., Ltd., Hangzhou 311200, China; guojiafuture@foxmail.com

**Keywords:** *Helleborus* L., phenotypic traits, genetic diversity, Analytic Hierarchy Process (AHP), comprehensive evaluation, genetic resources

## Abstract

*Helleborus orientalis* L. is a valuable winter-flowering and understory landscape plant, but its application and breeding are hindered by poor heat tolerance and the lack of a robust germplasm evaluation system. In this study, 51 *Helleborus* L. hybrid individuals obtained through manual open pollination were evaluated using coefficient of variation (CV), Shannon–Weaver diversity index (H′), correlation analysis, principal component analysis (PCA), and cluster analysis to assess genetic diversity and ornamental value based on 17 phenotypic traits. The results showed rich phenotypic diversity among the hybrids. Quantitative traits showed CV ranging from 9.48% to 37.99% and H′ between 0.77 and 1.51, with flower count and leaf length being the most variable. Qualitative traits had H′ values from 0.52 to 1.55, with sepal color showing the highest diversity. Significant correlations were detected among heat tolerance, pest resistance, leaf and petiole length, as well as plant and flower form. PCA extracted six principal components accounting for 74.50% of cumulative variance. Cluster analysis classified the 51 germplasms into five groups. Using the AHP model, a comprehensive evaluation system was established, and 13 elite individuals were selected for variety rights application and characterization. This study provides a reference for establishing DUS test guidelines and advancing breeding and utilization of *Helleborus* L.

## 1. Introduction

*Helleborus* L. is a perennial evergreen herbaceous ornamental plant belonging to the *Ranunculaceae* family, comprising approximately 22 species native to Western Asia and Southeastern Europe [1]. Species in this genus exhibit strong adaptability to cool climates, characterized by notable shade tolerance and cold hardiness [2]. Blooming primarily in winter and early spring, they are prized as flowering groundcovers for understory planting due to their elegant flowers and evergreen foliage [3]. Additionally, *Helleborus* L. is also an endangered medicinal resource. Its rhizomes contain diverse bioactive compounds, demonstrating significant potential for pharmaceutical research and development [4]. Among these species, *H. orientalis* L. is highly valued in horticulture and widely cultivated as a high-quality potted plant and cut flower [5]. It is distinguished by its vibrant flower colors, ease of hybridization, vigorous growth, rapid development, drought tolerance at maturity, and low susceptibility to pests and diseases. Consequently, it has become a mainstream choice for commercial cultivation and low-maintenance garden landscaping across Europe, North America, and other regions worldwide.

Current research on *Helleborus* L. has advanced in areas such as garden cultivation [6], tissue culture [7], introduction and domestication, heat tolerance [8], and the pharmacology of rhizome extracts [9]. However, challenges persist in its practical application and breeding. Although existing *H. orientalis* L. hybrids demonstrate strong cold tolerance, most are sensitive to high temperature and humidity, which limits their large-scale cultivation in subtropical monsoon climate regions [8]. Meanwhile, market demand for ornamental traits of *Helleborus* L. continues to rise. There is an urgent need for novel cultivars featuring rare colors, unique sepal markings, double flower type, and more compact architectures suited to potted environments. Therefore, studying the ornamental traits and ecological adaptability of *Helleborus* L. hybrid progenies is crucial for cultivating and promoting superior germplasm.

Plant phenotypic traits are determined by both genetic background and environmental factors [10]. They are key indicators of ornamental value and are essential for parental selection and variety screening in hybrid breeding [11]. This has been confirmed in species such as *Pennisetum alopecuroides*, *Lagerstroemia indica*, and *Manihot esculenta* [12,13,14]. Phenotypic traits are closely related to genetic diversity. Therefore, analyzing these traits can provide valuable insights into germplasm genetic diversity and help identify germplasm with superior traits. Currently, genetic diversity is typically studied through morphological, cytological, and molecular marker methods [15]. Although SSR marker analysis has been conducted on young leaf tissues of *Helleborus* L. [2], the genetic basis of floral organs and plant phenotypes remains unclear. Moreover, the absence of unified ornamental evaluation standards and a dedicated Distinctness, Uniformity, and Stability (DUS) testing system for new *Helleborus* L. varieties has severely limited cultivar protection and commercial application. Thus, systematically elucidating the correlations among phenotypic traits and establishing a comprehensive ornamental value evaluation system for *Helleborus* L. are crucial for effective genetic resource utilization and breeding.

The Analytic Hierarchy Process (AHP) is a decision analysis method that combines qualitative judgment with quantitative calculation [16]. It breaks down complex problems into a hierarchical structure, using pairwise comparisons and matrix calculations to quantify subjective experience [17]. Currently, AHP has been successfully applied in evaluating plants such as *Rhododendron* [18], *Calcareimontana* [19], and *Cornus* spp. [20] and selecting superior varieties of *Camellia* spp. [21]. Since *Helleborus* L. is an important ornamental flowering plant, qualitative traits such as flower color, type, sepal stripe pattern, and flower posture are more significant than quantitative traits such as flower diameter and bloom count in its evaluation. Thus, the AHP method, which combines qualitative and quantitative assessments, is well-suited for the comprehensive evaluation of *Helleborus* L. germplasm. Its application will provide a scientific approach for establishing a robust evaluation framework for *Helleborus* L. resources.

This study systematically evaluated phenotypic diversity and trait variation in 51 *Helleborus* L. hybrid individuals, aiming to address the research gap in phenotypic trait inheritance in this genus. Furthermore, this study established a comprehensive evaluation system for ornamental value to support the development of DUS testing guidelines and germplasm assessment.

## 2. Results

### 2.1. Analysis of Phenotypic Trait Diversity in Helleborus × hybridus L. Progeny

#### 2.1.1. Quantitative Trait Diversity

Frequency distribution histograms of seven quantitative traits (Figure 1) revealed that plant height (PHt), petiole length (PL), flower diameter (FD), and sepal length (SL) followed unimodal and approximately symmetric distributions, which did not significantly deviate from normality (*p* > 0.05, Shapiro–Wilk test). Their respective peaks were observed at approximately 33.00 cm, 17.00 cm, 6.50 cm, and 3.50 cm. In contrast, leaf blade length (LBL) significantly deviated from normality (*p* < 0.05). LBL was right-skewed, with a concentration of values between 17.00–21.00 cm and a broad spread toward higher values. Collectively, these distribution patterns suggest that traits such as PHt, PL, FD, and SL show high genetic stability in the hybrid progeny. Conversely, the skewed distributions of LBL imply that they may be influenced by environmental factors or major-effect genes.

To further investigate the variation patterns and genetic characteristics of quantitative traits, we analyzed the variation and genetic diversity of seven quantitative traits. As shown in Table 1, all traits displayed a wide range of variation, with coefficients of variation (CV) ranging from 9.48% to 37.99%. Among them, TFC displayed the highest CV (37.99%), followed by PSFC (35.04%), while SL showed the lowest (9.48%). The Shannon–Weaver diversity index (H′) revealed that LBL had the highest diversity (H′ = 1.51), followed by TFC (H′ = 1.35) and PSFC (H′ = 1.29), whereas SL had the lowest (H′ = 1.09). Notably, substantial phenotypic variation was observed in PHt and TFC. The difference between the tallest and shortest individuals in PHt was 28.60 cm, and the maximum TFC reached 46, which was 2.26 times the mean value. These findings demonstrate clear differences in both phenotypic variability (CV) and genetic diversity (H′) among the quantitative traits across the 51 hybrid progenies. Floral abundance-related traits such as TFC and PSFC exhibited high CV and H′ values, indicating strong potential for selecting high-yielding individuals. In contrast, SL and FD displayed low CV and H′ values, reflecting high genetic conservation and phenotypic stability. LBL, though moderately variable, showed the highest H′ value, indicating rich underlying genetic diversity. Moreover, the extreme phenotypes identified in this population—such as a PL of 45.60 cm and a TFC of 46—constitute valuable breeding resources for the genetic improvement of *Helleborus* L.

#### 2.1.2. Qualitative Trait Diversity

Frequency distribution analysis of ten qualitative traits (Table 2) indicated that flower type (FT) and curvature of sepal edges (CSE) were highly skewed in their distribution: 84.31% of individuals displayed the double flower type, and 76.47% exhibited flat sepal edges. Moderate distributional skewness was observed in plant habit (PlH), flower bearing posture (FBP), and growth vigor (GV), whereas sprouting ability (SA), heat resistance (HR), and disease and pest resistance (DPR) showed relatively balanced distributions. The main color of the sepal upper surface (MC) was highly variable, with white (33.33%) and purple (31.37%) being the most frequent, followed by pink (17.65%). Rare colors such as green and orange each accounted for less than 2%. Regarding sepal stripe pattern type (SSPT), the absence of patterns was most common (43.14%), stellate patterns were not observed, and the remaining types were distributed relatively evenly.

The Shannon–Weaver diversity index (H′) was calculated for each qualitative trait based on frequency distribution data (Table 2). The H′ values ranged from 0.52 to 1.55, reflecting considerable variation in genetic diversity across traits. Sepal-related traits generally exhibited relatively high diversity, with MC showing the highest value (H′ = 1.55), followed by SSPT (H′ = 1.31) and CSE (H′ = 1.08). SA, GV, and DPR displayed moderate diversity, whereas PlH and HR were comparatively low. The lowest diversity indices were observed for FBP (H′ = 0.66) and FT (H′ = 0.52).

In summary, qualitative traits in *Helleborus × hybridus* L. progeny displayed complex segregation and significant genetic differentiation. Sepal-related traits such as MC and SSPT showed high genetic diversity, while FT, FBP, and CSE (with flat edges accounting for 76.47%) exhibited high genetic conservation. Rare colors and semi-involute phenotypes occurred at very low frequencies, indicating considerable challenges for breeding improvement.

### 2.2. Correlation Analysis of Phenotypic Traits in Helleborus × hybridus L. Progeny

Spearman correlation analysis was performed on 17 phenotypic traits of 51 hybrid individuals. The results showed varying degrees of correlation among these traits, with some showing significant associations. Among all 136 trait pairs analyzed, 32 pairs (23.5%) showed significant correlations, with 26 of these pairs (19.1% of the total) having an absolute correlation coefficient |ρ| ≥ 0.3 (Figure 2).

Specifically, HR and DPR showed the highest correlation (ρ = 0.77), followed by LBL and PL (ρ = 0.69), as well as GV and DPR (ρ = 0.67). PlH was highly significantly positively correlated with FT, but significantly negatively correlated with SA, PL, and CSE. SA was significantly positively correlated with LBL and PL but not significantly correlated with other traits. Both PSFC and TFC were highly significantly positively correlated with PHt and GV. In addition, TFC was also significantly positively correlated with HR and negatively correlated with SL. SSPT was significantly positively correlated with CSE, HR, and DPR. Meanwhile, CSE was also significantly positively correlated with GV. Overall, phenotypic traits in the *Helleborus × hybridus* L. progeny were predominantly positively correlated, with few negative correlations. However, the proportion of strong correlations (|ρ| ≥ 0.3) was relatively low (19.10%), indicating a considerable degree of genetic independence among traits.

### 2.3. Principal Component Analysis of Phenotypic Traits in Helleborus × hybridus L. Progeny

Principal component analysis (PCA) was conducted on 17 phenotypic traits of 51 hybrid individuals (Table 3). Based on the criterion of eigenvalues greater than 1, six principal components (PCs) were extracted, with eigenvalues of 3.41, 2.58, 2.34, 1.68, 1.41, and 1.25, and variance contribution rates of 20.06%, 15.19%, 13.77%, 9.86%, 8.26%, and 7.36%, respectively. The cumulative contribution rate reached 74.50%, indicating that these six PCs retained most of the information from the original traits and effectively explained the main characteristics of the 17 phenotypic traits in the hybrid progeny.

PC1 (20.06% variance) was heavily loaded with GV, HR, and DPR, reflecting overall resistance performance. PC2 (15.19%) was associated with PHt, PSFC, and TFC, representing growth and floral productivity. PC3 (13.77%) was linked to SA, LBL, and PL, indicating vegetative growth capacity. PC4 (9.86%) was dominated by FD and SL, representing floral organ size. PC5 (8.26%) was primarily characterized by CSE, while PC6 (7.36%) showed high loadings on FT and MC, corresponding to floral morphology.

In the three-dimensional space defined by the first three PCs, the 51 hybrids showed a dispersed distribution (Figure 3), indicating clear phenotypic segregation among progeny. Overall, these results demonstrate substantial phenotypic variation in the hybrid population. Notably, DPR, HR, GV, PHt, TFC, and PSFC were identified as key traits underlying phenotypic diversity, providing important criteria for germplasm evaluation and cultivar breeding.

### 2.4. Cluster Analysis of Helleborus × hybridus L. Progeny

R-type and Q-type cluster analyses were performed using average linkage hierarchical clustering based on Euclidean distance, for the 17 phenotypic traits and the 51 hybrid individuals, respectively (Figure 4 and Figure 5).

The R-type clustering results (Figure 4) showed that at a Euclidean distance of 20, the 17 phenotypic traits were divided into five major clusters. Cluster I included HR, DPR, GV, SSPT, and FBP, which are associated with stress resistance and ecological adaptability. Cluster II consisted of FD, SL, LBL, PL, and SA, reflecting characteristics of nutritional and reproductive organs. Cluster III contained only CSE. Cluster IV comprised FT and MC, representing floral traits. Cluster V included PSFC, TFC, PlH, and PHt, related to flower production and plant conformation. Notably, within Cluster I, HR and DPR clustered together first, indicating the strongest correlation between the two traits, followed by the clustering of HR and GV. In Cluster II, LBL and PL were grouped initially, followed by FD and SL, which was highly consistent with the previous correlation analysis (Figure 2). The R-type clustering results revealed groups of traits with coordinated variation among specific phenotypic trait combinations. However, most other traits showed larger Euclidean distances and dispersed clustering suggest relatively independent evolution. These traits can effectively distinguish germplasm and are suitable for the classification and evaluation studies of *Helleborus* L. genetic resources.

The Q-type clustering results showed that at a Euclidean distance of 15, 51 hybrid individuals were divided into five major clusters (Figure 5):Cluster I contained 41 hybrids, characterized by superior plant height (PHt; mean: 32.90 cm), larger flower diameter (FD; mean: 6.22 cm), and higher total flower count per plant (TFC; mean: 21.50), as shown in Table 4. The majority of individuals exhibited a dispersive plant habit (PlH; 58.54%), double flowers (FT; 85.37%), and flat sepal edges (CSE; 73.17%). However, this cluster generally displayed intermediate performance in ecological adaptability traits, as summarized in Table 5. Representative individuals were H3 and H39.Cluster II comprised two hybrids, distinguished by the flower diameter (FD; mean: 8.03 cm) and a high number of flowers per stem (PSFC; mean: 4.17), as confirmed in Table 4. All individuals in this cluster showed double flowers (FT; 100%) and sepal edges with margined spotting patterns (SSPT; 100%). In addition, they were rated as strong in ecological adaptability (Table 5).Cluster III contained two hybrids, which exhibited high plant height (PHt; mean: 32.60 cm), along with longer leaf blade length (LBL; mean: 20.10 cm), petiole length (PL; mean: 16.55 cm), and sepal length (SL; mean: 4.50 cm), as supported by Table 4. All individuals displayed single flowers (FT; 100%) and horizontal flower bearing posture (FBP; 100%). Traits related to ecological adaptability, including GV, HR, and DPR, were generally rated as medium, as detailed in Table 5.Cluster IV included five hybrids characterized by intermediate plant height (PHt; mean: 28.61 cm) and relatively short leaf blade length (LBL; mean: 10.97 cm) and petiole length (PL; mean: 12.03 cm), as indicated in Table 4. All plants had double flowers (FT; 100%), with yellow as the predominant main color of sepal upper surface (MC; 60%). This cluster was consistently rated as poor in ecological adaptability (Table 5).Cluster V contained one hybrid, which showed greater plant height (PHt; mean: 32.00 cm) and a high total flower count (TFC; mean: 21.00), according to Table 4. This individual displayed black as the main color of sepal upper surface (MC; 100%) and semi-involute sepal edges (CSE; 100%). It also showed strong performance in ecological adaptability, as summarized in Table 5.

Overall, the *Helleborus × hybridus* L. individuals exhibited a relatively dispersed structure with indistinct subpopulation differentiation, consistent with the R-type clustering results which showed that most traits did not form strong, cohesive clusters, as evidenced by the long branches and dispersed distribution in the dendrogram (Figure 4).

### 2.5. Comprehensive Evaluation of Ornamental Value of Helleborus × hybridus L. Progeny

#### 2.5.1. Construction of Judgment Matrix and Consistency Test

Experts were invited to evaluate the ornamental value indicators of *Helleborus × hybridus* L. progeny. The Saaty 1–9 scale method (Table 6) was used to conduct pairwise comparisons of the evaluation indicators within the same level, with independent scoring based on the relative importance of each indicator.

Based on the expert scoring results, four judgment matrices were constructed: Target Layer A-Criteria Layer (A-B_1_-B_3_), Criteria Layer B_1_-Index Layer (B_1_-C_1_-C_5_), Criteria Layer B_2_-Index Layer (B_2_-C_6_-C_14_), and Criteria Layer B_3_-Indicator Layer (B_3_-C_15_-C_17_) (Table 7). The consistency ratio (CR) of each judgment matrix was calculated using yaahp 10.1 software. When CR < 0.1, the judgment matrix was considered to have acceptable consistency, and the results were valid. The results showed that all four judgment matrices had CR values below 0.1, passing the consistency test.

#### 2.5.2. Ranking of Phenotypic Trait Weights

The judgment matrices that passed the consistency test were subjected to weighted calculation using Excel 2019 to determine the weight values (*Wi*) of each element in the criteria layer (B_1_–B_3_) and the index layer (C_1_–C_17_) (Table 8).

Results of weight analysis showed that among the criteria layers, flower traits (B_2_) dominated with a weight of 0.60, indicating their greatest influence on the ornamental value of *Helleborus × hybridus* L. progeny; ecological adaptability (B_3_) was of secondary importance, while plant traits (B_1_) had the lowest weight. Within the index layer, TFC (C_9_) had the highest total weight (0.19) making it the primary selection indicator in *Helleborus* L. hybrid breeding; HR (C_16_) ranked second (0.17). In addition, PSFC (C_8_) and FBP (C_10_) also had relatively high comprehensive weights, at 0.12 and 0.10, respectively, whereas indicators such as LBL (C_4_), PL (C_5_), and FD (C_6_) had relatively low comprehensive weights (Figure 6).

Overall, in the comprehensive evaluation system for ornamental value of *Helleborus × hybridus* L. progeny, the weight of flower trait indicators was significantly higher than that of other trait indicators, indicating that flower-related traits are the decisive factors in ornamental value assessment. This weight distribution reflects the biological characteristics of *Helleborus* L. as an ornamental flowering plant and provides a reference for subsequent germplasm evaluation and selection of superior individuals.

#### 2.5.3. Comprehensive Evaluation of Hybrid Progeny

According to the scoring criteria, the 51 hybrid individuals were quantitatively evaluated for each trait.

Combined with the total weight values (*Wi*) of each evaluation indicator in Table 8, the comprehensive score (S) of each individual was calculated using the weighted summation formula:(1)$$S=\sum_{i=1}^{17}W_{i}\times X_{i}$$

(S represents the comprehensive evaluation score of the individual, *Wi* is the comprehensive weight value of the *i*-th trait, and *Xi* is its quantitative value).

The results (Table 9) showed that the comprehensive scores of the 51 individuals ranged from 3.08 to 4.36. Among them, H31 received the highest score, while H2 had the lowest. Based on the comprehensive scores, the hybrid progeny were classified into three grades: Grade I (S ≥ 4.0) included 13 individuals with outstanding ornamental value, representing core candidate materials for new cultivar development, such as H31, H3, and H39; Grade II (4.0 > S ≥ 3.5) consisted of 29 individuals with good ornamental value, suitable as foundational material for breeding improvement, including representatives H43, H20, and H51; Grade III (S < 3.5) contained 9 individuals with relatively poor comprehensive ornamental performance, which require careful selection in hybrid breeding, exemplified by H11, H16, and H34. The detailed scores for each trait per individual are provided in Appendix A.

Based on the comprehensive evaluation of phenotypic traits, the top three superior germplasms were H31 (score: 4.36), H3 (4.26), and H39 (4.25), which exhibited outstanding performance in multiple traits including plant height (PHt), sprouting ability (SA), flower diameter (FD), total flower count (TFC), and ecological adaptability. In contrast, the three lowest-ranked germplasms—H2 (3.08), H19 (3.10), and H23 (3.26)—were characterized by relatively short plant height, small flower diameter, absence of sepal spots, and poor ecological adaptability. Detailed phenotypic values for all individuals are provided in Appendix A.

Statistical analysis revealed that in 51 hybrid individuals, Grade I superior individuals accounted for 25.49%, Grade II for 56.86%, and Grade III for 17.65%. The combined proportion of superior and promising individuals (Grade I + Grade II) reached 82.35%, indicating significant improvement in ornamental traits within this hybrid population. These results demonstrate abundant favorable genetic resources, providing valuable germplasm for subsequent breeding of elite cultivars and selection of hybrid parents in *Helleborus* L.

### 2.6. Phenotypic Characterization of Elite Selections of Helleborus × hybridus L. Progeny

Based on comprehensive evaluation scores, 13 Grade I *Helleborus × hybridus* L. individuals were selected for variety rights application, and their phenotypic traits were characterized in detail (Figure 7). At the plant trait level, these superior accessions predominantly exhibited a dispersive plant habit, with H39 and H47 showing a bushy form. Plant height ranged from 30.00 to 45.00 cm, suitable for group planting in landscapes. They also displayed strong sprouting ability, promoting rapid clump formation and enhancing horticultural value. Additionally, these selections showed well-expanded leaves and long petioles, further contributing to their ornamental superiority.

Flower traits clearly highlighted the core ornamental advantages of these elite individuals. The average flower diameter reached 6.41 cm, with H14 attaining a maximum of 8.20 cm. All 13 accessions were double-flowered with rich petal layers and dense blooming, resulting in outstanding visual appeal. Flowers were primarily horizontally oriented, improving upon the drooping habit of parental plants. Sepal colors were mainly pink, white, or yellow, some with margined patterning, and relatively long sepals with curved edges enhanced esthetic quality.

In terms of ecological adaptability, the superior selections exhibited vigorous growth and demonstrated breakthroughs in heat tolerance through cultivation practice. Notably, H3 has become the first *Helleborus* L. cultivar capable of large-scale open-field summer cultivation in regions south of the Yangtze River.

In summary, these elite *Helleborus × hybridus* L. progeny display stable growth, outstanding floral traits, and high comprehensive ornamental value, making them promising candidates for winter and understory landscaping in the lower Yangtze River region of China.

## 3. Discussion

### 3.1. Variation and Diversity of Phenotypic Traits of Helleborus × hybridus L. Progeny

Plant phenotypic traits are determined by genetic factors and environmental influences [22,23]. Phenotypic diversity analysis can quantify the degree of variation and dispersion of traits, reveal patterns of variation and genetic differentiation within populations, and thereby provide a basis for mining and selecting superior germplasm resources [11]. Quantitative traits are typically controlled by polygenes, exhibit continuous variation, and are easily affected by the environment. Qualitative traits, on the other hand, are regulated by single or oligogenes and show typical discontinuous distribution [24].

This study analyzed 17 phenotypic traits in 51 manually open-pollinated *Helleborus × hybridus* progeny, revealing extensive variation and high diversity across traits. Among quantitative traits, TFC (CV = 37.99%, H′ = 1.35) and PSFC (CV = 36.04%, H′ = 1.29) showed the highest variation and genetic diversity (Table 1). For qualitative traits, MC (H′ = 1.55) and SSPT (H′ = 1.31) were the most diverse (Table 2). Notably, most qualitative traits exhibited higher H′ values than quantitative traits (except FT and FBP), suggesting richer discrete variation in this population. These findings align with previous studies on *Chrysanthemum morifolium* leaf traits [25] and *Jasminum sambac* genetic diversity [26]. Based on these results, traits such as TFC and LBL—showing both high CV and high heritability—will be prioritized as selection indicators in subsequent breeding. This strategy is expected to accelerate the development of new *Helleborus* L. varieties with improved floral abundance and leaf morphology.

### 3.2. Correlation Analysis of Phenotypic Traits

Phenotypic trait analysis is widely used to study the classification and genetic diversity of ornamental plants, as demonstrated in species such as *Chimonanthus praecox* [27], *Rhododendron* [18], and *Xanthoceras sorbifolium* [28]. Correlation analysis among phenotypic traits helps reveal inter-trait relationships, enabling indirect selection to improve efficiency and accelerate breeding progress [29]. In this study, correlation analysis uncovered complex association patterns among 17 phenotypic traits in *Helleborus × hybridus* L. progeny, with particularly notable relationships between floral and growth traits. PSFC and TFC were highly significantly positively correlated with PHt and GV (Figure 2). FT was highly significantly positively correlated with PlH, and SL with LBL, indicating coordinated regulation between floral organ development and vegetative growth. Conversely, significant negative correlations were observed between PlH and SA, PL, and CSE, implying a potential developmental trade-off during morphogenesis. These association patterns provide valuable references for breeding and establish a foundation for future molecular marker techniques (such as SSR or SNP).

PCA reduced the 17 phenotypic traits to six principal components, collectively explaining 74.50% of the total variance (Table 3). PC1–PC3 primarily represent vegetative growth capacity, ecological adaptability, and flower production traits, offering selection criteria for high-yield and stress-tolerant varieties. PC4–PC6 mainly reflect floral organ traits, providing a basis for selecting improved flower forms and colors. This approach simplifies the evaluation of ornamental traits and facilitates early selection in breeding, as successfully applied in *Rosa* [30] and *Nymphaea* [31]. Q-type cluster analysis classified the 51 hybrids into five distinct groups (Figure 5), revealing clear phenotypic differentiation. PlH, PHt, TFC, FT, MC, SSPT, and ecological adaptability were identified as key traits for distinguishing germplasm resources. This classification provides diverse parental materials for breeding and supports improved breeding efficiency.

### 3.3. Comprehensive Evaluation of Ornamental Value of Helleborus × hybridus L. Progeny Based on AHP

The Analytic Hierarchy Process (AHP) is a decision-making method that combines subjective judgment with quantitative analysis, enabling the calculation of factor weights through pairwise comparisons and judgment matrices [32]. It has been widely applied in germplasm evaluation of plants such as *Iris* [33], *Rhododendron* [18] and *Impatiens oxyanthera* [34]. As germplasm resources form the material basis for breeding new varieties, systematic evaluation is crucial for ornamental plant research [35]. Comprehensive evaluation of ornamental plants typically involves multiple aspects such as plant morphology, leaf characteristics, flower traits, and ecological adaptability [36]. As a typical flowering ornamental plant, the ornamental value of *Helleborus × hybridus* L. is primarily determined by flower traits, which play a decisive role in its evaluation. Since the observed indicators in this study were dominated by floral traits, and qualitative traits significantly outnumbered quantitative traits, the AHP method was more suitable than Principal Component Analysis (PCA) for the comprehensive evaluation of ornamental value of *Helleborus × hybridus* L. progeny.

Given that floral traits dominated the observed indicators and qualitative traits significantly outnumbered quantitative ones in this study, AHP was more suitable than PCA for comprehensively evaluating the ornamental value of *Helleborus × hybridus* progeny. For the first time, 51 hybrid individuals were systematically assessed from a horticultural perspective using an AHP model comprising 3 criterion layers and 17 indicators. Among the criterion layers, flower traits (B_2_) had the highest weight (0.60), and among the indicators, total flower count (C_9_; 0.19) and heat tolerance (C_16_; 0.17) ranked highest (Table 8), indicating that floral performance and adaptation to high-temperature environments are key factors determining ornamental value. Based on comprehensive scores, the hybrids were classified into three grades (Table 9), with 13 Grade I elite individuals selected for variety rights application, facilitating the breeding and commercialization of new cultivars. This evaluation highlights the dominant role of floral traits in ornamental quality. As an evergreen perennial, *Helleborus* L. also exhibits significant ornamental potential in leaf morphology and color. Future efforts could focus on improving leaf traits such as venation patterns and coloration to enhance landscape performance during non-flowering periods and broaden breeding applications.

## 4. Materials and Methods

### 4.1. Experimental Conditions

The test materials were cultivated in the *Helleborus* L. germplasm nursery of Zhejiang Institute of Landscape Plants and Flowers (30°04′06.94″ N, 120°13′39.17″ E). Located in a subtropical monsoon climate at 17 m elevation, the site has an average annual temperature of 17.8 °C, 1435 mm of precipitation, a frost-free period of 230–260 days, and relative humidity between 74% and 85%. The nursery was shaded 75% from May to September annually. Climatic conditions during the growing period are shown in Figure 8. The flat planting area has well-established drainage, irrigation, and ventilation systems. To assess natural phenotypic traits and resistance levels under natural climatic conditions and pest stress, no artificial irrigation, fungicides, or insecticides were used during the experimental period.

### 4.2. Plant Materials

Select horticultural varieties such as ‘Allen Purple’, ‘Black Swan’, and ‘White Primadress’, with excellent ornamental characteristics as hybrid parents (Figure 9 and Table 10). The parental materials were purchased from Zhejiang Licai Horticulture Co., Ltd. (Hangzhou, China) (‘Allen Purple’, ‘Emotions Anemone Picotee’, ‘Tutu’, ‘Ruby’, ‘Petal Princess’, ‘Golden Jade’, ‘Bella’, ‘Mint Princess’), Kunming Lakeland Agricultural Technology Co., Ltd. (Kunming, China) (‘Quintessa Pink’, ‘Candy’), and Yunnan Yingmao Modern Agriculture Co., Ltd. (Kunming, China) (‘Black Swan’, ‘Peach Primadress’, ‘White Primadress’, ‘Elegance Honey Wine’, ‘Apricot Gelato’, ‘Ellen Green’).

Open manual pollination was conducted during the flowering period in March 2019 to obtain hybrid seeds. These hybrid seeds were sown in October 2019, germinated in January 2020, repotted in April of the same year, and transplanted to the germplasm nursery in December 2020. Initial flowering occurred from February to April 2021. From over 2000 hybrid progeny that survived after sowing, 51 hybrid individuals with stable phenotypic traits and excellent ornamental performance over two consecutive years were preliminarily selected. These were propagated by division in September–October 2022 (the fourth autumn after sowing). Each divided clump retained 3–5 buds and was planted with a spacing of 30 × 50 cm. Systematic observation and recording of the phenotypic traits of the hybrid individuals were conducted in the third year after natural flowering following division (i.e., in February 2025, from the sixth winter to the seventh spring after sowing).

### 4.3. Selection and Measurement of Phenotypic Traits

Since there is no unified evaluation standard for the ornamental value of *Helleborus* L. genus nor an international UPOV test guideline for new varieties, this study referred to the DUS test guidelines for *Helleborus* L. in Japan [37]. Based on its system of 38 standardized trait observations, combined with evaluation standards for related plants such as *Paeonia lactiflora* [38] and *Clematis* [39] and relevant expert experience, 17 key ornamental traits including plant habit, flower diameter, and main color of sepal upper surface were selected as observation indicators. During the full blooming period, three plants from each hybrid line were randomly selected for systematic measurement. The complete list of phenotypic parameters with their abbreviations and measurement units is provided in Appendix A.

Specific measurement methods were as follows: plant height (PHt), leaf blade length (LBL), and petiole length (PL) were measured using a tape measure; flower diameter (FD) and sepal length (SL) were measured with a vernier caliper. Plant habit (PlH) was determined by the height-to-width ratio: >1 for erect, =1 for bushy, and <1 for dispersive. Flower type (FT) was classified based on the number of petals or the degree of nectary petalization. The main color of sepal upper surface (MC) was identified using the RHS Color Chart (Royal Horticultural Society, 2015). Flower bearing posture (FBP) was visually recorded: horizontal (parallel to pedicel), downward (<90°), or semi-erect (>90°). Sepal stripe pattern type (SSPT) and curvature of sepal edges (CSE) were documented visually. The primary stem flower count (PSFC) referred to the total number of flowers on the earliest flowering stalk, while the total flower count (TFC) was the sum of the flower counts of all main flower stalks. Sprouting ability (SA) and growth vigor (GV) were evaluated from December to February, while heat tolerance (HR) and disease and pest resistance (DPR) were assessed in September over three consecutive years. SA was graded by rhizome bud count: bad (<3 buds), medium (4–8 buds), or strong (>8 buds). HR was graded based on damage: strong (minimal damage, occasional leaf scorch), medium (<30% leaf margin scorching), or bad (>30% leaf scorching or burn). DPR was graded: strong (no damage), medium (<30% leaf spots from single pest/disease), or bad (>50% leaf spots with >30% rhizome rot from multiple pests/diseases). The phenotypic data for all 51 hybrid individuals, including detailed measurements for all 17 traits (Table 11), were recorded and are provided in Appendix A.

### 4.4. Coefficient of Variation and Shannon–Weaver Diversity Index

The coefficient of variation (CV) was used to evaluate the phenotypic variability of seven quantitative traits, including plant height, leaf blade length, petiole length, flower diameter, primary stem flower count, total flower count, and sepal length. The Shannon–Weaver diversity index (H′) was applied to analyze the genetic diversity of all 17 phenotypic traits. Prior to calculation, both quantitative and qualitative traits were preprocessed [27]. Continuous data of quantitative traits were discretized using equal-width binning, categorizing the 51 hybrid individuals into five grades. Discrete data of qualitative traits were grouped, and the frequency distribution of each grade was calculated (Table 2).

The formula for the coefficient of variation (CV) is:(2)$$CV=\frac{\sigma }{\mu }\times 100\%$$
where *σ* is the standard deviation and *μ* is the mean.

The formula for the Shannon–Weaver diversity index (H′) is:(3)$$H^{\prime }=-\sum_{i=1}^{k}p_{i}\ln(p_{i})$$
where *H*′ is the genetic diversity index, *k* is the number of grades, and *p_i_* is the frequency of the phenotypic trait in the *i*-th grade.

### 4.5. Correlation Analysis and Principal Component Analysis

The original data of quantitative traits were standardized using Z-Score normalization, while qualitative traits were assigned quantitative values according to the grading rules outlined in Table 12. Based on the preprocessed data, correlation analysis and PCA were performed using SPSS 27.0 software [27]. Correlation analysis was conducted using Spearman’s rank correlation method to generate a correlation coefficient matrix among traits, with the significance level set at *p* < 0.05. Principal components were extracted based on eigenvalues (λ > 1). The component loadings were then used to interpret the principal components. Eigenvalues, variance contribution rates, and cumulative variance contribution rates were calculated. The suitability for factor analysis was assessed using the KMO measure (0.61; see Appendix A for individual MSA values) and Bartlett’s test of sphericity (*p* < 0.001), which confirmed that the variables were sufficiently correlated for the analysis.

### 4.6. Construction of AHP Analysis and Evaluation Model

Based on 17 key phenotypic traits, the AHP model was established to develop a comprehensive evaluation system for *Helleborus* L. phenotypes. This system quantifies trait weights through expert decision-making, enabling systematic hierarchical evaluation of germplasm resources. According to the growth characteristics, ornamental traits of *Helleborus* L., and expert opinions, the comprehensive evaluation system was divided into three hierarchical levels. The first level is the Target Layer (A), which represents the comprehensive evaluation of ornamental value in *Helleborus × hybridus* L. progeny. The second level is the Criteria Layer (B), comprising three first-level indicators: plant traits (B1), flower traits (B2), and ecological adaptability (B3). The third level is the Index Layer (C), which includes 17 specific phenotypic traits such as plant habit (C1), plant height (C2), flower diameter (C6), flower type (C7), growth vigor (C15), heat tolerance (C16), and disease and pest resistance (C17) (Table 13). Yaaph 10.1 software was used to construct judgment matrices and perform consistency tests (CR < 0.1).

Through consultations with experts and referencing evaluation systems for ornamental plants such as *Rhododendron* [18], *Paeonia lactiflora* [40], and *Phalaenopsis Aphrodite* [41], scoring criteria for the 17 evaluation indicators were established based on the growth characteristics and phenotypic observation data of *Helleborus × hybridus* L. (Table 14). Each indicator was classified into five grades according to measured data and assigned a score from 1 to 5, with higher scores indicating better trait performance. During the scoring phase, five experts were invited to conduct on-site evaluations independently and rate each indicator based on landscape ornamental value. To minimize subjective bias, the arithmetic mean of the scores for each indicator was used as the final evaluation result.

### 4.7. Data Processing and Analysis

Microsoft Excel 2019 was used to calculate the maximum, minimum, mean, standard deviation, variance, and coefficient of variation for quantitative traits, as well as the distribution frequency and Shannon–Weaver diversity index (H′) for qualitative traits. Spearman correlation analysis, PCA, and cluster analysis were performed using SPSS 27 software [42]. Finally, frequency distribution histograms for quantitative traits, heatmaps of correlation coefficients, three-dimensional PCA scatter plots, and weight distribution charts were generated using Origin 2024 software.

## 5. Conclusions

In summary, the 51 *Helleborus* L. hybrid individuals displayed rich phenotypic diversity, including extreme phenotypes with high breeding potential. Analysis of trait variation revealed that total flower count (TFC), primary stem flower count (PSFC), main sepal color of sepal upper surface (MC), and sepal stripe pattern type (SSPT) showed particularly high diversity, offering a genetic basis for selecting high-yield individuals and novel floral traits. Correlation analysis indicated the strongest association between heat tolerance (HR) and disease and pest resistance (DPR). Although most traits were positively correlated, the low proportion of strong correlations suggested relative trait independence. PCA extracted six principal components explaining 74.50% of total variance, identifying disease and pest resistance (DPR), heat tolerance (HR), and growth vigor (GV) as key drivers of phenotypic variation. Q-type cluster analysis classified the hybrids into five distinct groups. The AHP-based evaluation confirmed the predominant role of floral traits—particularly total flower count, primary stem flower count, and flower bearing posture—in determining ornamental value. The superior individuals selected through comprehensive scoring provide valuable germplasm resources for future *Helleborus* L. breeding programs.

## Figures and Tables

**Figure 1 plants-14-03226-f001:**
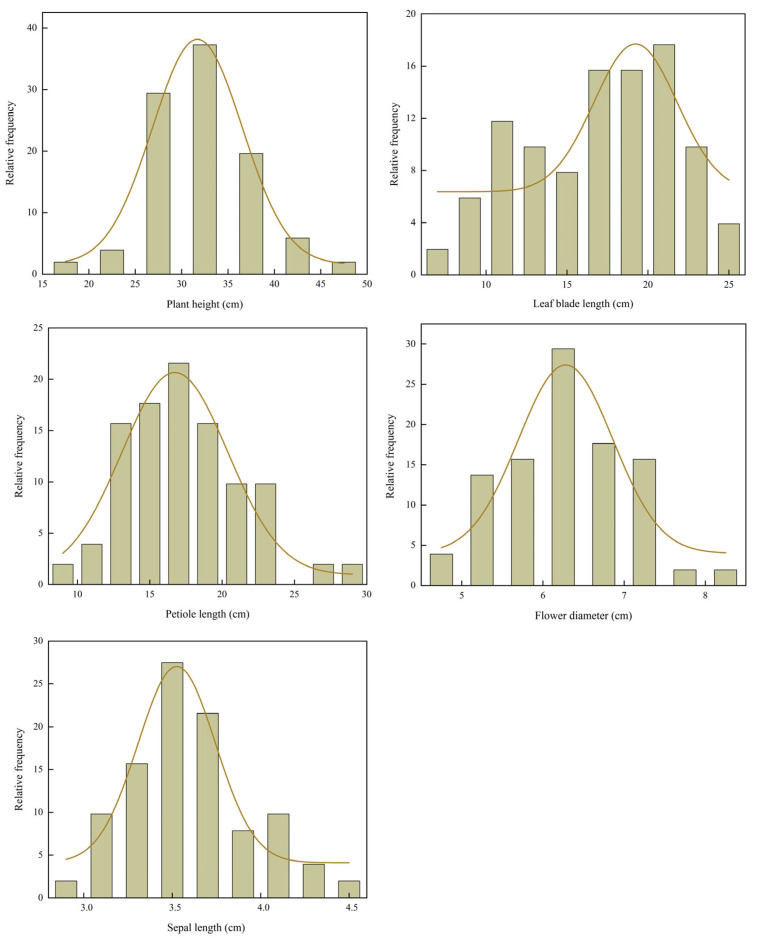
Frequency distribution histograms of quantitative traits of *Helleborus × hybridus* L. progeny. (The lines in the frequency distribution histogram represent the distribution curves).

**Figure 2 plants-14-03226-f002:**
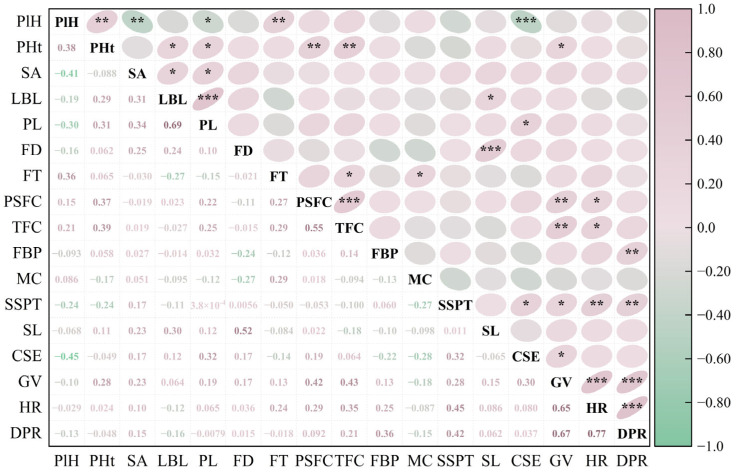
Correlation heat map of phenotypic traits of *Helleborus × hybridus* L. progeny (Significance levels: * *p* < 0.05, ** *p* < 0.01, *** *p* < 0.001).

**Figure 3 plants-14-03226-f003:**
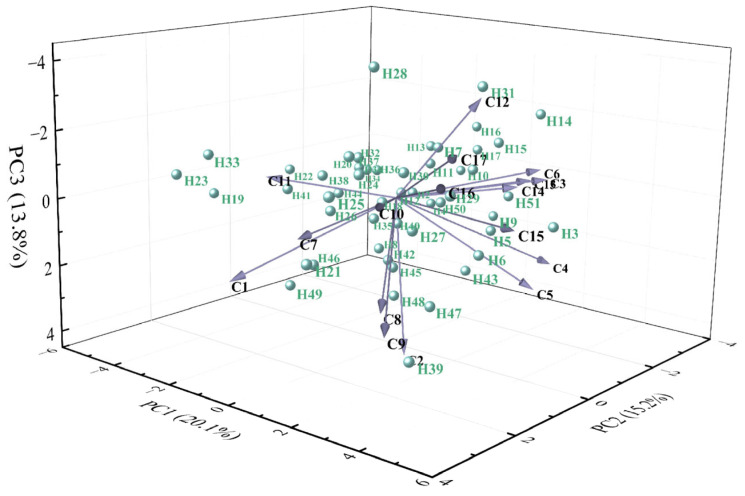
Phenotypic distribution of *Helleborus × hybridus* L. progeny in PCA space (PC1/PC2/PC3) (PCA was performed based on 17 phenotypic traits, with the cumulative contribution rate of PC1-PC3 being 49.1%; blue scatter points represent different sample individuals, and arrows correspond to different phenotypic trait variables).

**Figure 4 plants-14-03226-f004:**
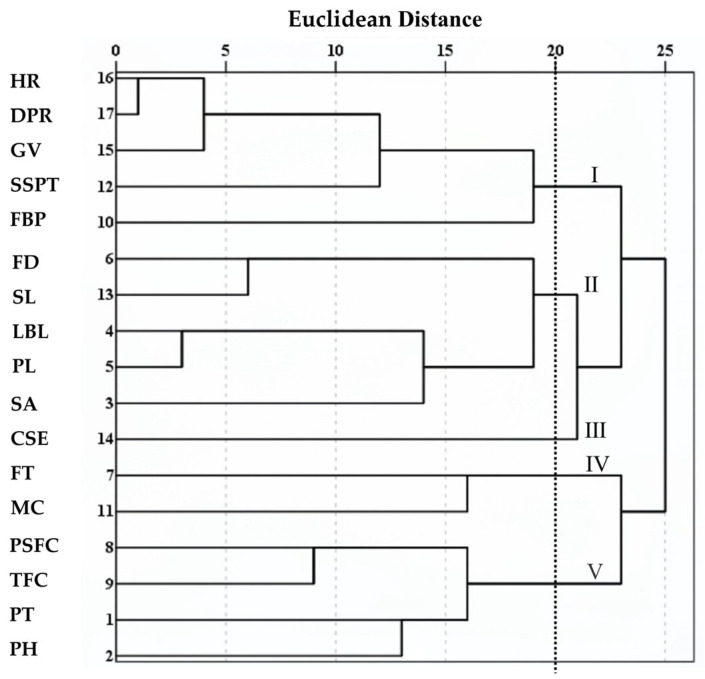
R-cluster analysis of 17 phenotypic traits of *Helleborus × hybridus* L. progeny.

**Figure 5 plants-14-03226-f005:**
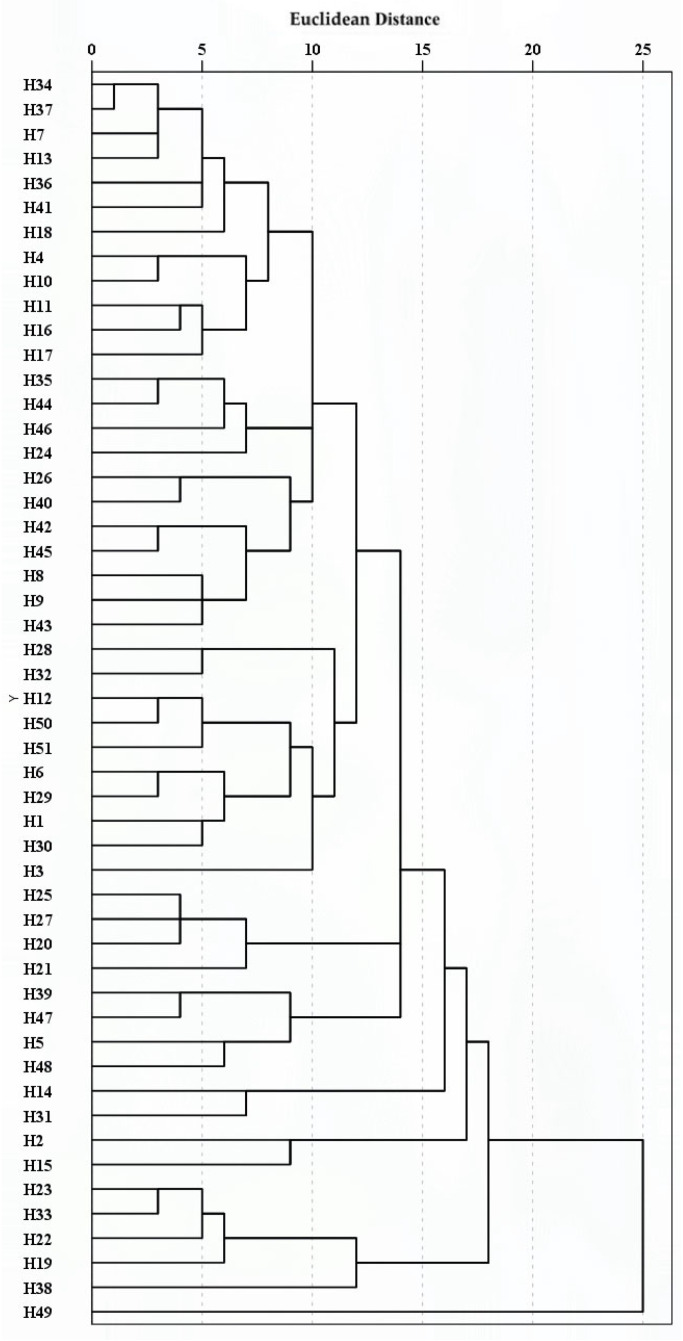
Q-cluster analysis of 51 *Helleborus* L. hybrid individuals.

**Figure 6 plants-14-03226-f006:**
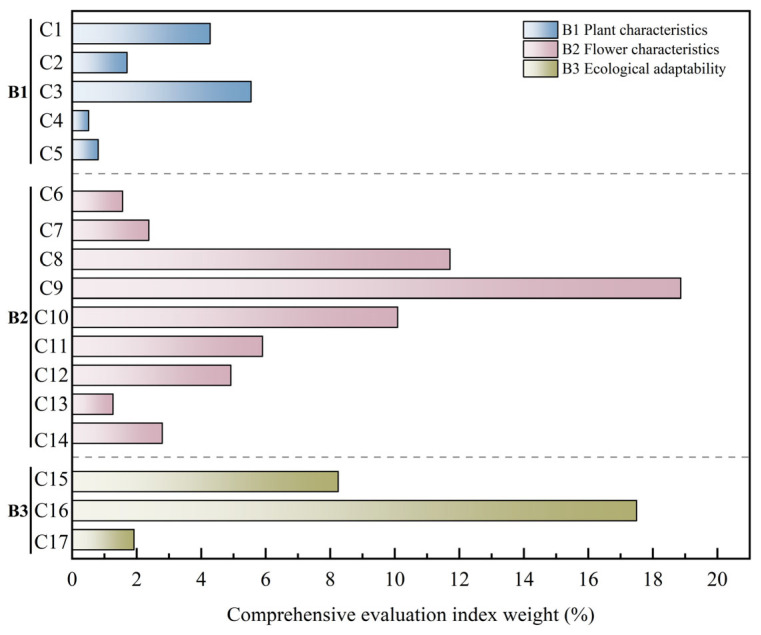
Weights of comprehensive evaluation indicators of *Helleborus × hybridus* L. progeny.

**Figure 7 plants-14-03226-f007:**
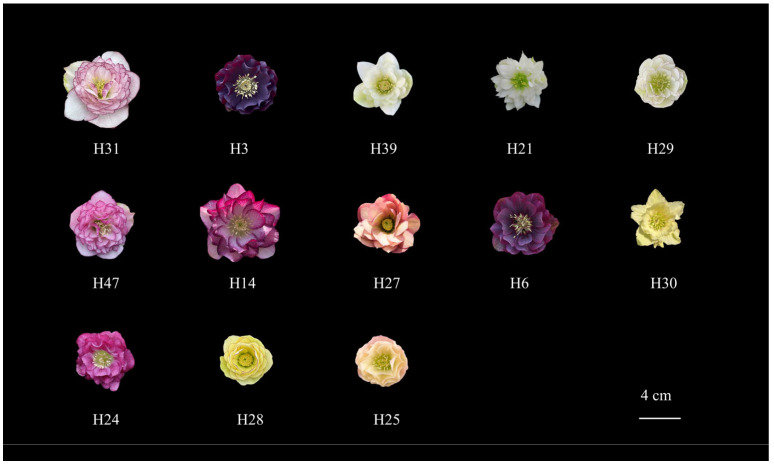
Selected superior individuals of *Helleborus × hybridus* L. progeny.

**Figure 8 plants-14-03226-f008:**
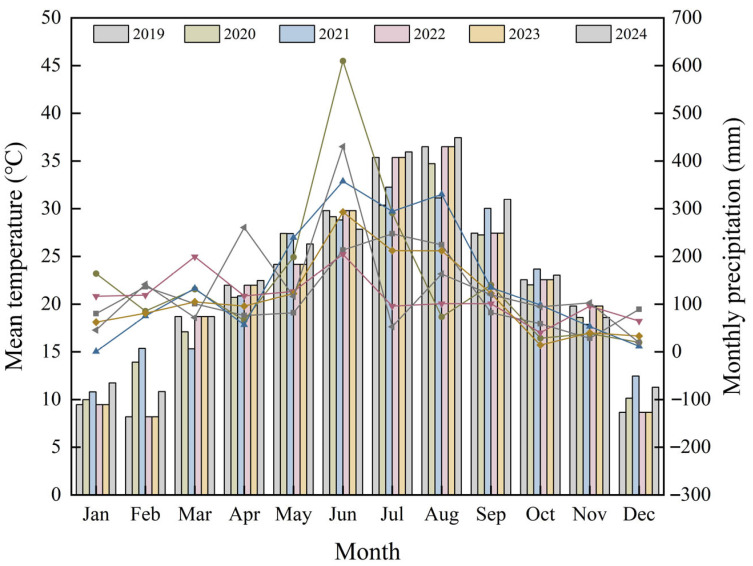
Monthly mean temperature and precipitation conditions from 2019 to 2024.

**Figure 9 plants-14-03226-f009:**
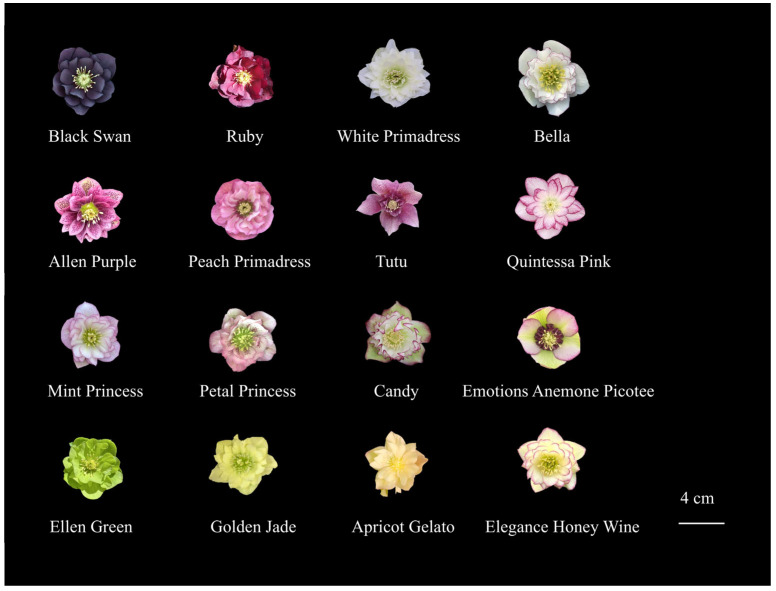
Phenotype of *Helleborus × hybridus* L. hybridization.

**Table 1 plants-14-03226-t001:** Analysis of variation and diversity in quantitative traits of *Helleborus × hybridus* L. progeny.

Traits	Max	Min	Mean ± SD	Range	CV (%)	H′
PHt (cm)	45.60	17.00	32.07 ± 5.19	17.00–47.60	16.18	1.04
LBL (cm)	24.00	7.40	16.79 ± 4.51	7.40–24.00	26.87	1.51
PL (cm)	28.00	9.30	17.17 ± 3.87	9.30–28.00	22.53	1.26
FD (cm)	8.20	4.86	6.20 ± 0.72	4.86–8.20	11.58	0.94
PSFC	8.00	3.00	4.65 ± 1.63	3–6	35.04	1.29
TFC	46.00	7.00	20.33 ± 7.73	7–46	37.99	1.35
SL (cm)	4.40	2.90	3.58 ± 0.34	2.90–4.40	9.48	0.77

**Table 2 plants-14-03226-t002:** Analysis of variation and diversity in qualitative traits of *Helleborus × hybridus* L. progeny.

Traits	Hybrid F_1_ Generation Trait Grading and Segregation Ratio (%)							H′
	1	2	3	4	5	6	7	
PlH	56.86	13.73	29.41					0.95
SA	13.73	45.10	41.18					1.00
FT	3.92	11.76	84.31					0.52
FBP	37.25	62.75	0					0.66
MC	33.33	31.37	17.65	1.96	1.96	9.80	3.92	1.55
SSPT	43.14	19.61	19.61	0	17.65			1.31
CSE	76.47	21.57	1.96					1.08
GV	19.61	35.29	45.10					1.04
HR	13.73	52.94	33.33					0.92
DPR	11.76	52.94	35.29					0.96

**Table 3 plants-14-03226-t003:** Loading matrix of PCA for phenotypic traits of *Helleborus × hybridus* L. progeny.

Traits	Principal Component						Total Load
	1	2	3	4	5	6	
PlH	−0.18	0.48	−0.50	0.08	−0.43	0.20	−0.35
PHt	−0.11	0.83	0.09	0.10	−0.16	−0.22	0.53
SA	0.19	−0.20	0.70	0.25	−0.02	0.14	1.06
LBL	−0.22	0.210	0.78	0.21	−0.02	−0.25	0.71
PL	0.01	0.37	0.77	−0.07	0.18	−0.09	1.17
FD	0.11	−0.04	0.07	0.84	0.11	−0.12	0.97
FT	0.09	0.26	−0.20	0.15	−0.01	0.74	1.03
PSFC	0.15	0.61	0.10	−0.22	0.35	0.27	1.26
TFC	0.33	0.75	0.07	−0.13	0.04	0.11	1.17
FBP	0.42	0.04	0.16	−0.49	−0.42	−0.29	−0.58
MC	−0.12	−0.25	0.04	−0.33	−0.17	0.76	−0.07
SSPT	0.60	−0.33	−0.08	0.06	0.32	−0.12	0.45
SL	0.10	−0.05	0.23	0.80	−0.13	0.00	0.95
CSE	0.09	0.03	0.12	0.03	0.91	−0.12	1.06
GV	0.74	0.36	0.12	0.21	0.20	−0.03	1.60
HR	0.87	0.16	0.01	0.05	0.03	0.20	1.32
DPR	0.91	−0.01	−0.01	−0.010	−0.11	−0.08	0.69

**Table 4 plants-14-03226-t004:** Descriptive statistics of quantitative traits in *Helleborus × hybridus* L. progeny clusters.

Group		Cluster I	Cluster II	Cluster III	Cluster IV	Cluster V
**PHt (cm)**	Mean	32.90	23.15	32.60	28.61	32.00
	CV (%)	15.84	10.08	6.07	9.59	-
**LBL (cm)**	Mean	17.27	15.94	20.10	10.97	21.00
	CV (%)	24.31	58.20	3.52	23.01	-
**PL (cm)**	Mean	17.83	14.87	16.55	12.03	22.00
	CV (%)	20.76	17.40	11.54	19.18	-
**FD (cm)**	Mean	6.22	8.03	5.55	5.82	4.86
	CV (%)	9.37	2.99	6.37	13.46	-
**PSFC**	Mean	4.78	4.17	3.55	3.41	2.90
	CV (%)	37.05	4.41	5.98	14.81	-
**TFC**	Mean	21.50	15	10.50	16.72	21.00
	CV (%)	37.31	37.71	47.14	19.56	-
**SL (cm)**	Mean	3.59	4.50	4.50	3.88	11.00
	CV (%)	8.20	15.71	47.14	19.49	-

**Table 5 plants-14-03226-t005:** Frequency distribution of qualitative traits in *Helleborus × hybridus* L. progeny clusters.

Group		Cluster I	Cluster II	Cluster III	Cluster IV	Cluster V
**PlH**	**Mode**	Dispersive	Dispersive	Dispersive	Dispersive	Dispersive
	**Frequency** (%)	58.54	100	100	80	100
**SA**	**Mode**	Strong	Strong or Medium	Medium	Medium	Strong
	**Frequency** (%)	48.78	50	100	60	100
**FT**	**Mode**	Double	Double	Single	Double	Double
	**Frequency** (%)	85.37	100	100	100	100
**FBP**	**Mode**	Horizontal	Horizontal or Downward	Horizontal	Horizontal	Horizontal
	**Frequency** (%)	65.85	50	100	65.85	100
**MC**	**Mode**	White or Purple	White or Purple	White	Yellow	Black
	**Frequency** (%)	34.15	50	100	60	100
**SSPT**	**Mode**	Absent	Margined	Absent or Veined	Absent	Absent
	**Frequency** (%)	39.02	100	50	80	100
**CSE**	**Mode**	Flat	Flat	Flat	Flat	Semi-involute
	**Frequency** (%)	73.17	100	100	100	100
**GV**	**Mode**	Strong	Strong	Medium or Strong	Bad	Bad
	**Frequency** (%)	51.22	100	50	80	100
**HR**	**Mode**	Medium	Strong	Bad or Strong	Bad or Medium	Medium
	**Frequency** (%)	63.41	100	50	40	100
**DPR**	**Mode**	Medium	Strong	Medium or Strong	Medium	Bad
	**Frequency** (%)	53.66	100	50	80	100

**Table 6 plants-14-03226-t006:** 1–9 Ratio scaling method.

Significance Scale	Significance of Corresponding Scale
1	Indicates that two elements are of equal importance
3	Indicates moderate importance of one element over another
5	Indicates strong importance of one element over another
7	Indicates very strong importance of one element over another
9	Indicates extreme importance of one element over another
2, 4, 6, 8	Intermediate values between two adjacent judgments
Reciprocal of scale	If element *i* has one of the above non-zero numbers assigned to it when compared to element *j*, then *j* has the reciprocal value when compared to *i*

**Table 7 plants-14-03226-t007:** Judgment matrix and consistency test.

Hierarchical Model	Judgment Matrix										Consistency Test
A-Bi	A	B1	B2	B3							λ = 3.0055; CR = 0.0053
	B1	1	1/5	1/2							
	B2	5	1	2							
	B3	2	1/2	1							
B1-Ci	B1	C1	C2	C3	C4	C5					λ = 5.4029; CR = 0.0899
	C1	1	5	1/2	7	7					
	C2	1/5	1	1/6	6	3					
	C3	2	6	1	7	5					
	C4	1/7	1/6	1/7	1	1/2					
	C5	1/7	1/3	1/5	2	1					
B2-Ci	B2	C6	C7	C8	C9	C10	C11	C12	C13	C14	λ = 10.0708; CR = 0.0917
	C6	1	1/2	1/7	1/6	1/6	1/6	1/6	2	1/3	
	C7	2	1	1/6	1/6	1/6	1/3	1/3	4	1/2	
	C8	7	6	1	1/2	1	4	4	6	7	
	C9	6	6	2	1	4	7	7	6	6	
	C10	6	6	1	1/4	1	2	4	6	7	
	C11	6	3	1/4	1/7	1/2	1	1	7	5	
	C12	6	3	1/4	1/7	1/4	1	1	5	3	
	C13	1/2	1/4	1/6	1/6	1/6	1/7	1/5	1	1/4	
	C14	3	2	1/7	1/6	1/7	1/5	1/3	4	1	
B3-Ci	B3	C15	C16	C17							λ = 3.1013; CR = 0.0974
	C15	1	1/3	6							
	C16	3	1	7							
	C17	1/6	1/7	1							

**Table 8 plants-14-03226-t008:** Weights and ranking of evaluation criteria.

Target Layer	Criteria Layer	Sub-Weight Value (*Wi*)	Index Layer	Sub-Weight Value (*Wi*)	Total Weight Value (*Wi*)	Ranking
Comprehensive evaluation of ornamental value of *Helleborus × hybridus* L. progeny	Plant characteristics (B1)	0.1285	C1 PHt	0.3329	0.0428	9
			C2 PlH	0.1326	0.0170	13
			C3 SA	0.4318	0.0555	7
			C4 LBL	0.0398	0.0051	17
			C5 PL	0.0629	0.0081	16
	Flower characteristics (B2)	0.5949	C6 FD	0.0263	0.0156	14
			C7 FT	0.04	0.0238	11
			C8 PSFC	0.1969	0.1171	3
			C9 TFC	0.3171	0.1886	1
			C10 FBP	0.1696	0.1009	4
			C11 MC	0.0992	0.0590	6
			C12 SSPT	0.0827	0.0492	8
			C13 SL	0.0213	0.0127	15
			C14 CSE	0.047	0.0280	10
	Ecological adaptability (B3)	0.2766	C15 GV	0.2981	0.0825	5
			C16 HR	0.6325	0.1749	2
			C17 DPR	0.0694	0.0192	12

**Table 9 plants-14-03226-t009:** Evaluation and ranking of composite scores for ornamental value of 51 *Helleborus × hybridus* L. individuals.

Ranking	Cultivar Number	B1 Score	B2 Score	B3 Score	Composite Score	Grade	Ranking	Cultivar Number	B1 Score	B2 Score	B3 Score	Composite Score	Grade
1	H31	0.39	2.59	1.38	4.36	I	27	H35	0.49	2.17	1.11	3.77	II
2	H3	0.47	2.40	1.38	4.26	I	28	H32	0.41	2.05	1.30	3.76	II
3	H39	0.63	2.24	1.38	4.25	I	29	H5	0.48	2.05	1.21	3.74	II
4	H21	0.39	2.46	1.38	4.23	I	30	H46	0.47	2.12	1.11	3.69	II
5	H29	0.47	2.36	1.38	4.21	I	31	H13	0.41	2.17	1.11	3.69	II
6	H47	0.60	2.20	1.38	4.18	I	32	H42	0.49	1.98	1.19	3.65	II
7	H14	0.37	2.43	1.38	4.18	I	33	H15	0.45	1.81	1.38	3.65	II
8	H27	0.39	2.40	1.38	4.18	I	34	H48	0.48	2.05	1.11	3.64	II
9	H6	0.48	2.33	1.36	4.18	I	35	H12	0.43	2.06	1.11	3.60	II
10	H30	0.45	2.30	1.38	4.14	I	36	H44	0.45	2.12	1.02	3.60	II
11	H24	0.42	2.25	1.38	4.04	I	37	H37	0.36	2.11	1.11	3.57	II
12	H28	0.39	2.26	1.38	4.03	I	38	H36	0.42	2.02	1.11	3.55	II
13	H25	0.31	2.34	1.38	4.03	I	39	H41	0.42	2.01	1.11	3.53	II
14	H43	0.50	2.30	1.19	3.99	II	40	H49	0.37	2.16	1.00	3.53	II
15	H20	0.35	2.22	1.38	3.95	II	41	H10	0.42	2.26	0.83	3.51	II
16	H51	0.47	2.27	1.19	3.93	II	42	H17	0.42	2.05	1.02	3.50	II
17	H40	0.49	2.19	1.19	3.87	II	43	H11	0.42	1.96	1.11	3.49	III
18	H1	0.45	2.19	1.21	3.85	II	44	H16	0.37	2.09	1.02	3.48	III
19	H18	0.41	2.25	1.19	3.85	II	45	H34	0.35	2.00	1.11	3.45	III
20	H38	0.37	2.19	1.28	3.84	II	46	H33	0.37	1.95	1.02	3.35	III
21	H9	0.52	2.23	1.09	3.84	II	47	H4	0.41	2.09	0.83	3.33	III
22	H45	0.51	2.14	1.19	3.84	II	48	H22	0.40	2.04	0.85	3.29	III
23	H8	0.54	2.21	1.09	3.83	II	49	H23	0.31	1.93	1.02	3.26	III
24	H7	0.42	2.28	1.13	3.82	II	50	H19	0.36	1.92	0.83	3.10	III
25	H50	0.43	2.17	1.19	3.79	II	51	H2	0.40	1.76	0.93	3.08	III
26	H26	0.41	2.26	1.11	3.78	II							

**Table 10 plants-14-03226-t010:** Key traits of parental lines in *Helleborus × hybridus* L. hybridization.

Variety Name	Main Color of Sepal Upper Surface	Sepal Stripe Pattern Type	Flower Type
‘Black Swan’	Black	Absent	Double
‘Ruby’	Purple Red	Absent	Double
‘White Primadress’	White	Absent	Double
‘Bella’	White	Margined	Double
‘Allen Purple’	Purple	Dusted	Double
‘Peach Primadress’	Dark pink	Mottled	Double
‘Tutu’	Purple	Dusted	Semi-double
‘Quintessa Pink’	Pink	Veined	Double
‘Mint Princess’	Pink	Margined	Double
‘Petal Princess’	Pink	Dusted	Double
‘Candy’	Pink	Veined	Double
‘Emotions Anemone Picotee’	Dark pink	Margined	Single
‘Ellen Green’	Green	Absent	Double
‘Golden Jade’	Yellow	Absent	Double
‘Apricot Gelato’	Orange	Absent	Double
‘Elegance Honey Wine’	Light Yellow	Margined	Double

**Table 11 plants-14-03226-t011:** Characterization of 17 qualitative and quantitative traits.

Traits	Abbreviation	Unit	Category
Plant habit	PlH	-	qualitative
Plant height	PHt	cm	quantitative
Sprouting ability	SA	-	qualitative
Leaf blade length	LBL	cm	quantitative
Petiole length	PL	cm	quantitative
Flower diameter	FD	cm	quantitative
Flower type	FT	-	qualitative
Primary stem flower count	PSFC	count	quantitative
Total flower count	TFC	count	quantitative
Flower bearing posture	FBP	-	qualitative
Main color of sepal upper surface	MC	-	qualitative
Sepal stripe pattern type	SSPT	-	qualitative
Sepal length	SL	cm	quantitative
Curvature of sepal edges	CSE	-	qualitative
Growth vigor	GV	-	qualitative
Heat resistance	HR	-	qualitative
Disease and pest resistance	DPR	-	qualitative

**Table 12 plants-14-03226-t012:** Grouping description of qualitative traits of *Helleborus × hybridus* L. progeny.

Group	Qualitative Traits									
	PlH	SA	FT	FBP	MC	SSPT	CSE	GV	HR	DPR
1	Dispersive	Bad	Single	Downward	White	Absent	Flat	Bad	Bad	Bad
2	Erect	Medium	Semi-double	Horizontal	Purple	Dusted/mottled	Semi-involute	Medium	Medium	Medium
3	Bushy	Strong	Double	Semi-erect	Pink	Veined/radial	Involute	Strong	Strong	Strong
4					Green	Stellate				
5					Orange	Margined				
6					Yellow					
7					Black					

**Table 13 plants-14-03226-t013:** Hierarchical structure of comprehensive evaluation of ornamental traits of *Helleborus × hybridus* L. progeny.

Target Layer (A)	Criteria Layer (B)	Index Layer (C)
Comprehensive evaluation of ornamental value of *Helleborus × hybridus* L. progeny	Leaf characteristics (B1)	C1 Plant habit
		C2 Plant height
		C3 Sprouting ability
		C4 Leaf blade length
		C5 Petiole length
	Flower characteristics (B2)	C6 Flower diameter
		C7 Flower type
		C8 Primary stem flower count
		C9 Total flower count
		C10 Flower bearing posture
		C11 Main color of sepal upper surface
		C12 Sepal stripe pattern type
		C13 Sepal length
		C14 Curvature of sepal edges
	Ecological adaptability (B3)	C15 Growth vigor
		C16 Heat resistance
		C17 Disease and pest resistance

**Table 14 plants-14-03226-t014:** Evaluation criteria and scoring system for key ornamental traits of *Helleborus × hybridus* L. progeny.

Traits	Score				
	5	4	3	2	1
C1 PlH	Bushy	Erect	Dispersive		
C2 PHt (cm)	>45	40–45	35–40	30–35	<30
C3 SA	Strong	Medium	Bad		
C4 LBL (cm)	>20	16–20	12–16	8–12	<8
C5 PL (cm)	>20	18–20	16–18	14–16	<14
C6 FD (cm)	>8	7–8	5–6	3–4	<3
C7 FT	Double	Semi-double	Single		
C8 PSFC	>10	8–9	6–7	4–5	<4
C9 TFC	>35	25–35	15–24	5–15	<5
C10 FBP	Semi-erect	Horizontal	Downward		
C11 MC	Rare color (black)	Solid colors (yellow, orange, green)	Solid colors (purple-red, pink)	Solid color (purple)	White
C12 SSPT	Margined	Stellate	Veined or radial	Dusted or mottled	Absent
C13 SL	>4.5	4–4.5	3.5–4	3–3.5	<3
C14 CSE	Involute	Semi-involute	Flat		
C15 GV	Strong	Medium	Bad		
C16 HR	Strong	Medium	Bad		
C17 DPR	Strong	Medium	Bad		

## Data Availability

The data provided in this study are available upon request from the corresponding author. Due to privacy concerns, these data are not publicly available.

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
