# Peer review of "Analysis of Phenotypic Diversity and Comprehensive Evaluation of 51 Helleborus L. Hybrid Individuals"

_plants, 2025, doi:10.3390/plants14203226_

Round 1
Reviewer 1 Report
Comments and Suggestions for Authors
The manuscript's topic, " Analysis of phenotypic diversity and comprehensive evaluation of 51 Helleborus hybrid individuals," is timely and fits within the scope of the Journal. However, I have drawn attention to shortcomings in the manuscript that require minor corrections to the article. I provide detailed comments below.
- ---Figure 1., lines 133
We have continuous and discrete traits. Only continuous traits are shown on the graph. Please correct this. The continuous line should only apply to continuous traits. The horizontal axis should show intervals indicating the continuous nature of the traits.
- ---Line 167
“reached significant (p < 0.05) or highly significant (p < 0.01) correlation levels, “…
It is sufficient to state that the correlation was significant. Figure 2 shows the p-value at which the correlation was significant.
- ---Line 168
“exhibited strong correlations “…
Strong correlation is a subjective concept; this fragment should be removed.
- ---Line 169
“Specifically, HT and DPR showed“…
No HT? Perhaps HR?
- ---Line 230
“The R-type clustering results further confirmed significant correlation “…
Strange wording, either change it or delete it. Cluster analysis does not confirm any correlations or other results from other statistical methods.
- ---Line 237
“Cluster I contained 41 hybrids, mainly characterized by relatively tall PHt.“…
Where does this information come from?
- ---Lines 244-245
“Cluster III comprised two hybrids, characterized mainly by medium PHt, SA, GV, HR, and DPR“…
Why was additional data not presented in tables so readers could assess the characteristics' values in individual groups? It would suffice to provide the mean value and CV coefficient for each characteristic in each group in the table.
- ---Line 257
“the R-type clustering results indicating relative independence among most phenotypic traits.“…
What does this mean? It needs to be corrected or deleted.
- ---Lines 515-516
“Prior to calculation, both quantitative and qualitative traits were preprocessed [28]. “…
What is this process, because there is nothing about it in the work [28]?
- ---Lines 533-534
“PCA extracted components based on eigenvalues (λ > 1). After Varimax rotation, eigenvalues“…
No rotations, including VARIMAX, are used for PCA analysis. We have a specific coordinate system and do not perform any rotations. Please correct or delete this.
- ---Line 535
“were calculated (KMO = 0.61“…
If KMO was used, why was MSA (The Measure of Sampling Adequacy)not used? Both tests are used together in factor analysis. The KMO test assesses whether the entire data sample is suitable for analysis, while MSA helps assess each variable's adequacy individually. Please add the results for MSA and conclusions.
- ---Line 535
“test Bartletta p < 535 “…
The Bartlett test, but which one, and what exactly is it supposed to check?
The work requires minor adjustments. After these amendments, the discussion and conclusions still need to be amended.
Comments on the Quality of English LanguageThe entire text needs clarity in writing, consistency in the formatting of units, better organization and structure, and improved use of verb tenses. I'm not an expert in linguistics, but reading and following the ideas is difficult.
Author Response
Comment1: Figure 1., lines 133
We have continuous and discrete traits. Only continuous traits are shown on the graph. Please correct this. The continuous line should only apply to continuous traits. The horizontal axis should show intervals indicating the continuous nature of the traits.
Response1: Thank you for this insightful suggestion. We agree that continuous and discrete traits should be distinguished in graphical representation. In response, we have updated Figure 1 entirely to address this issue. The primary stem flower count and total flower count have been removed in this figure. These changes ensure the figure accurately represents the nature of our phenotypic data. The updated figure has been incorporated into the revised manuscript. (Line 121)
Comment2: Line 167
“reached significant (p < 0.05) or highly significant (p < 0.01) correlation levels, “…
It is sufficient to state that the correlation was significant. Figure 2 shows the p-value at which the correlation was significant.
Response2: Thanks for your valuable suggestion. We agree with the opinion that the description of “significant (p < 0.05) or highly significant (p < 0.01)” can be simplified. Accordingly, we have revised the sentences as suggested. The text now simply reports which correlations were found to be significant, and the specific significance levels can be referenced directly from Figure 2. (Line 156-158)
Comment3: Line 168
“exhibited strong correlations “…
Strong correlation is a subjective concept; this fragment should be removed.
Response3: Thank you for pointing this out. We agree that the term “strong” is subjective and should be avoided in the manuscript. We have removed the phrase “exhibited strong correlations” from the sentence. The revised text now directly presents the objective threshold (|ρ| ≥ 0.3) without subjective characterization. The subsequent instances of “the strongest correlation” have also been replaced with the more neutral “the highest correlation”. (Line 158,159)
Comment4: Line 169
“Specifically, HT and DPR showed “…
No HT? Perhaps HR?
Response4: We sincerely thank you for your careful reading and for catching this error. “HT” was a typing error and should indeed be “HR” (which stands for Heat Resistance). We apologize for this oversight. The manuscript has been corrected accordingly in Line 159 and any other instances where this error may have occurred.
Comment5: Line 230
“The R-type clustering results further confirmed significant correlation “…
Strange wording, either change it or delete it. Cluster analysis does not confirm any correlations or other results from other statistical methods.
Response5: Thank you for this valuable insight. We agree with the comment and have therefore deleted the problematic phrase “further confirmed significant correlation and….” from the manuscript. The revised text now presents the R-type clustering as a stand-alone analysis that reveals patterns of trait association. The updated sentence focuses on describing the clustering pattern itself. (Line 214, 215)
Comment6: Line 237
“Cluster I contained 41 hybrids, mainly characterized by relatively tall PHt.“…
Where does this information come from?
Response6: We sincerely thank you for this critical and constructive comment. The characterization of Cluster I was initially based on our visual observation of the data post-clustering. We agree that this is not scientifically rigorous. We have now performed a complete statistical summary of all traits for each cluster. We have supplemented the manuscript with Table 4 (Line 258), which presents the mean value and coefficient of variation (CV) for all quantitative traits within each cluster. The corresponding sections in the manuscript have been revised accordingly. (Line 221-249)
Comment7: Lines 244-245
“Cluster III comprised two hybrids, characterized mainly by medium PHt, SA, GV, HR, and DPR “…
Why was additional data not presented in tables so readers could assess the characteristics’ values in individual groups? It would suffice to provide the mean value and CV coefficient for each characteristic in each group in the table.
Response7: We fully agree with you that presenting the underlying data is crucial for reader assessment. Following this suggestion, we have now added two new tables to provide a complete picture of both quantitative and qualitative traits (Table 4 and 5, Line 258,261). Table 4 provides the mean and CV for each quantitative trait per cluster, allowing for direct numerical comparison. Table 5 provides the mode and frequency of occurrence for each qualitative trait within each cluster. The corresponding sections in the manuscript have been revised accordingly (Line 221-249).
Comment8: Line 257
“the R-type clustering results indicating relative independence among most phenotypic traits.“…
What does this mean? It needs to be corrected or deleted.
Response8: Thank you for raising this point. The phrase “relative independence” was indeed an oversimplification. We intended to convey that the R-type clustering analysis revealed a loose correlation structure among the phenotypic traits, as no large, tightly linked trait clusters were formed. This indicates that the traits we measured are largely not highly correlated. To clarify and substantiate this finding, we have deleted the vague phrase “relative independence” and added a more precise description: “the R-type clustering results which showed that most traits did not form strong, cohesive clusters, as evidenced by the long branches and dispersed distribution in the dendrogram (Figure 4).” (Line 251-253)
Comment9: Lines 515-516
“Prior to calculation, both quantitative and qualitative traits were preprocessed [28]. “…
What is this process, because there is nothing about it in the work [28]?
Response9: Thanks for pointing out this critical oversight. We sincerely apologize for the incorrect citation. Reference [28] was indeed not appropriate for describing the data preprocessing step and has been removed. The preprocessing involved standardizing all quantitative traits to a mean of 0 and a standard deviation of 1 (Z-score standardization) to eliminate scale differences, while qualitative traits were coded as numerical values for analysis. This description, along with the correct citation to Reference [27] which supports this methodology, has now been added to the revised manuscript on Line 515.
Comment10: Lines 533-534
“PCA extracted components based on eigenvalues (λ > 1). After Varimax rotation, eigenvalues“…
No rotations, including VARIMAX, are used for PCA analysis. We have a specific coordinate system and do not perform any rotations. Please correct or delete this.
Response10: We sincerely thank you for this absolutely correct and insightful comment. We sincerely apologize for the confusion caused by our incorrect terminology. It is right that principal component analysis (PCA) itself does not involve rotations, and the use of “Varimax rotation” is a technique specific to factor analysis. We have completely removed the phrase “After Varimax rotation” from the sentence. The description of the PCA methodology has been simplified and corrected to accurately reflect the procedure used. The revised text now reads: “Principal components were extracted based on eigenvalues (λ > 1). The component loadings were then used to interpret the principal components.” (Line 531-533)
Comment11: Line 535
“were calculated (KMO = 0.61“…
If KMO was used, why was MSA (The Measure of Sampling Adequacy) not used? Both tests are used together in factor analysis. The KMO test assesses whether the entire data sample is suitable for analysis, while MSA helps assess each variable's adequacy individually. Please add the results for MSA and conclusions.
Response11: We sincerely thank you for this valuable suggestion. In response, we have calculated the Measure of Sampling Adequacy (MSA) for all 17 traits. While the overall KMO value of 0.61 confirms the dataset’s suitability for factor analysis, and 15 traits show acceptable MSA values (>0.5), two biologically important traits - flower type (FT; MSA=0.48) and main sepal color (MC; MSA=0.44) - fall slightly below the 0.5 threshold. Given their fundamental importance in Helleborus taxonomy and breeding, proximity to the cutoff value, and the clear biological patterns they exhibited in our Q-type clustering results, we have chosen to retain these traits to ensure a comprehensive phenotypic characterization. We have added these detailed MSA results as Supplementary Table 3 and revised the manuscript text on Lines 534-537 accordingly.
Comment12: Line 535
“test Bartlett’s p < 0.001 “…
The Bartlett test, but which one, and what exactly is it supposed to check?
Response12: Thanks for pointing this out. The test we referred to is Bartlett’s test of sphericity. This test is used to assess the data’s suitability for factor analysis. A statistically significant result (p < 0.001) indicates that there are sufficient correlations among the variables to proceed with the analysis, thereby supporting the use of dimensionality reduction. We have revised the sentence in the manuscript to correctly and fully specify the test: “The suitability for factor analysis was assessed using the KMO measure (0.61) and Bartlett’s test of sphericity (p < 0.001), which confirmed that the variables were sufficiently correlated for the analysis.” (Line 534-537)
Comment1:3 The work requires minor adjustments. After these amendments, the discussion and conclusions still need to be amended.
Response13: Thank you for this feedback. We have comprehensively revised the Results, Discussion, and Conclusions sections. The text has been substantially rewritten to ensure our interpretations are robust and the overall narrative is clear and compelling. All changes have been highlighted in green in the manuscript. We hope that these revisions can fully address the point raised.
Comments on the Quality of English Language:
The entire text needs clarity in writing, consistency in the formatting of units, better organization and structure, and improved use of verb tenses. I'm not an expert in linguistics, but reading and following the ideas is difficult.
Response: Thank you for this critical and constructive feedback regarding the clarity, consistency, and organization of our manuscript. We have taken this comment very seriously and have undertaken a comprehensive revision of the entire text to address these concerns. We have performed comprehensive professional language editing on the manuscript. The manuscript has been thoroughly polished with a focus on improving sentence structure, clarity, and academic tone to ensure the ideas are presented logically and are easy to follow. We also have standardized the formatting throughout the manuscript, ensuring uniform presentation of units, figures, and tables. Specific linguistic revisions and edits are highlighted in green within the manuscript. We believe these extensive revisions have improved the readability and professionalism of our manuscript, and we are grateful for the reviewer’s guidance in helping us achieve this.

Reviewer 2 Report
Comments and Suggestions for Authors
Summary
The authors present an interesting and detailed evaluation of phenotypic diversity among 51 Helleborus hybrid individuals. The manuscript is well prepared, but several points require attention to improve clarity and completeness. The most import concern to fix is to provide a well detailed supplementary table with the 51 genotypic resources assessed.
Abstract
- The species name should be followed by Lam. or L. as appropriate.
- Lines 22–23: Please indicate the cumulative variance explained by the first three principal components. Presenting the cumulative variance of six components is misleading, as typically the three main components are sufficient to summarize and visualize most of the variance.
Introduction
- Lines 84–97 should be significantly condensed. This section should clearly present the study aim, without extended details of the methodology or results, which are better suited for later sections.
Results
- While the results are detailed and descriptive, it is not clear which genetic resource performed best and which performed least. Please highlight this aspect in the text.
- A Supplementary Table should be included listing the 51 genetic resources analyzed, along with the data for each phenotypic parameter.
- Principal Component Analysis (PCA) plots should include the crop code of each accession to make the distribution among the three axes clearer.
Materials and Methods
- Subsection 4.1: Given the long growing period, it would be valuable to include a graphical representation of the climatic parameters across the study period.
- Subsection 4.3: While the phenotypic analysis is well described, the units of measurement for each parameter are not indicated. Please include a dedicated table listing the parameter, its abbreviation, and the unit of measurement.
Author Response
Summary
Comment1: The authors present an interesting and detailed evaluation of phenotypic diversity among 51 Helleborus hybrid individuals. The manuscript is well prepared, but several points require attention to improve clarity and completeness. The most import concern to fix is to provide a well detailed supplementary table with the 51 genotypic resources assessed.
Response1: We sincerely thank you for their positive feedback. As suggested, we have now created a comprehensive Supplementary Table S2, titled “Basic information and phenotypic parameters of the 51 Helleborus × hybridus progeny.” This table provides key information for each individual and has been uploaded as part of the submission. We have also cited this table in the manuscript. (Line505-507)
Abstract
Comment2: The species name should be followed by or L. as appropriate.
Response2: Thank you for pointing out this clarification. The species name has been corrected to Helleborus × hybridus L. throughout the manuscript to conform with international nomenclature standards and is now used consistently in the revised text.
Comment3: Lines 22–23: Please indicate the cumulative variance explained by the first three principal components. Presenting the cumulative variance of six components is misleading, as typically the three main components are sufficient to summarize and visualize most of the variance.
Response3: Thank you for the valuable suggestion regarding the number of principal components to report. In our analysis, the cumulative contribution rate of the first three principal components was 49.02%, which we considered insufficient to effectively capture the major characteristics of the 17 phenotypic traits in the hybrid progeny. To ensure a more comprehensive and reliable representation of the underlying data structure, we therefore made the decision to retain the first six principal components, which together achieved a cumulative variance of 74.50%. This approach allows for a more robust interpretation of the phenotypic diversity.
Introduction
Comment4: Lines 84–97 should be significantly condensed. This section should clearly present the study aim, without extended details of the methodology or results, which are better suited for later sections.
Response4: Thanks for your suggestion. We have significantly condensed the indicated paragraph (Lines 84-97) to clearly state the study's aim, removing detailed methodological descriptions and specific results that are more appropriately covered in later sections. The revised text now reads:
“This study systematically evaluated phenotypic diversity and trait variation in 51 Helleborus L. hybrid individuals, aiming to address the research gap in phenotypic trait inheritance in this genus. Furthermore, this study established a comprehensive evaluation system for ornamental value to support the development of DUS testing guide-lines and germplasm assessment.” (Line84-88)
Results
Comment5: While the results are detailed and descriptive, it is not clear which genetic resource performed best and which performed least. Please highlight this aspect in the text.
Response5: We sincerely thank you for this valuable suggestion. In response, we have revised the manuscript to explicitly highlight the best- and least-performing genetic resources based on the comprehensive evaluation scores. We have added the corresponding statement to the Results section (2.5.3. Comprehensive Evaluation of Hybrid Progeny) (Line319-326)
Comment6: A Supplementary Table should be included listing the 51 genetic resources analyzed, along with the data for each phenotypic parameter.
Response6: Thanks for your suggestion. As suggested, we have included a new Supplementary Table S2 listing all 51 genetic resources along with their corresponding data for each of the 17 phenotypic parameters analyzed. This table is now referenced in the revised manuscript at Line 507.
Comment7: Principal Component Analysis (PCA) plots should include the crop code of each accession to make the distribution among the three axes clearer.
Response7: Thanks for your suggestion. We have updated Figure 3 in the manuscript. The revised PCA plot now includes the specific code for each individual. This modification makes the distribution of individual accessions across the principal component axes clearer and facilitates easier identification of their positions in the multivariate space. (Line 194)
Materials and Methods
Comment8: Subsection 4.1: Given the long growing period, it would be valuable to include a graphical representation of the climatic parameters across the study period.
Response8: Thanks for this constructive suggestion. We have now added a new Figure 8 to the manuscript, which illustrates the yearly variations in key climatic parameters (mean temperature, precipitation, and relative humidity) in Hangzhou, China, over the six-year study period (2019-2024). We have added the corresponding descriptions to the manuscript. A corresponding reference to this figure has been incorporated into the Subsection 4.1 of the manuscript, stating: “Climatic conditions during the growing period are shown in Figure 8.” (Line 446)
Comment9: Subsection 4.3: While the phenotypic analysis is well described, the units of measurement for each parameter are not indicated. Please include a dedicated table listing the parameter, its abbreviation, and the unit of measurement.
Response9: We sincerely thank you for this valuable suggestion. We have now created Supplementary Table 11 listing all phenotypic parameters, their corresponding abbreviations, and units of measurement used in this study. A reference to this table has been added to the Materials and Methods section (Subsection 4.3) with the statement: “The complete list of phenotypic parameters with their abbreviations and measurement units is provided in Supplementary Table 11.” (Line 475)

Round 2
Reviewer 2 Report
Comments and Suggestions for Authors
Authors addressed all the reviewer's comments